# 3′-UTR Polymorphisms in Thymidylate Synthase with Colorectal Cancer Prevalence and Prognosis

**DOI:** 10.3390/jpm11060537

**Published:** 2021-06-09

**Authors:** Young-Joo Jeon, Sung-Hwan Cho, Eo-Jin Kim, Chang-Soo Ryu, Han-Sung Park, Jong-Woo Kim, Jeong-Yong Lee, Hui-Jeong An, Nam-Keun Kim

**Affiliations:** 1Department of Biomedical Science, College of Life Science, CHA University, Seongnam 13488, Korea; white0885@naver.com (Y.-J.J.); arana006@naver.com (S.-H.C.); regis2040@nate.com (C.-S.R.); hahnsung@naver.com (H.-S.P.); smilee3625@naver.com (J.-Y.L.); tody2209@naver.com (H.-J.A.); 2Division of Hematology/Oncology, Department of Internal Medicine, Kangbuk Samsung Hospital, Sungkyunkwan University School of Medicine, Seoul 06351, Korea; eojin1.kim@samsung.com; 3CHA Bundang Medical Center, Department of Surgery, CHA University, Seongnam 13496, Korea; 4College of Life Science, Gangneung-Wonju National University, 7 Jukheon-gil, Gangneung 25457, Korea

**Keywords:** thymidylate synthase, 3′-UTR, miRNA polymorphism, colorectal cancer, prevalence, prognosis

## Abstract

Colorectal cancer (CRC) is the third most common type of cancer and the second leading cause of cancer-related mortality in Western countries. Polymorphisms in one-carbon metabolism and angiogenesis-related genes have been shown to play important roles in tumor development, progression, and metastasis for many cancers, including CRC. Moreover, recent studies have reported that polymorphisms in specific microRNA (miRNA)-binding regions, which are located in the 3′-untranslated region (UTR) of miRNA-regulated genes, are present in a variety of cancers. Here, we investigated the association between two thymidylate synthase (*TYMS* or *TS)* 3′-UTR polymorphisms, 1100T>C [rs699517] and 1170A>G [rs2790], and CRC susceptibility and progression in Korean patients. A total of 450 CRC patients and 400 healthy controls were enrolled in this study, and genotyping at the TS locus was performed by polymerase chain reaction–restriction fragment length polymorphism (PCR-RFLP) or TaqMan allelic discrimination assays. We found that TS 1170A>G genotypes, as well as the TS 1100T-1170G and 1100C-1170A haplotypes, are strongly associated with CRC. The TS 1100TC+CC type was associated with a poor survival (OS and RFS) rate. In addition, levels of the TS 1100C and TS 1170G allele were found to be significantly increased in CRC tissue. Our study provides the first evidence for 3′-UTR variants in TS genes as potential biomarkers of CRC prognosis and prevention.

## 1. Introduction

Colorectal cancer (CRC) is the third most common cancer worldwide, with 1,849,518 new cases diagnosed per year and representing 10.2% of all new cancer cases. This disease is also the second most common cancer in women, with 823,303 cases per year, which represents 9.5% of total cancer cases in women [1]. The geographical incidence of CRC varies, but patterns are similar in men and women, and approximately 55% of CRC cases occur in more developed regions [2]. Several intrinsic factors, such as age, sex, diabetes mellitus, obesity, and inflammatory bowel disease, as well as extrinsic variables, including cigarette smoking, inadequate fiber intake, high alcohol consumption, red meat consumption, and a high-fat diet, are associated with increased CRC risk [3,4,5]. Thus, CRC susceptibility appears to be influenced by both genetic and environmental factors.

Recently, one-carbon (1C) metabolism has received considerable attention for its role in cancer malignancies, and consequently, therapeutic agents targeting this pathway are increasingly being developed as anticancer agents. Several genes are involved in 1C metabolism, including methylenetetrahydrofolate reductase (*MTHFR*), 5-methyltetrahydrofolate-homocysteine methyltransferase (*MTR*), 5-methyltetrahydrofolate-homocysteine methyltransferase reductase (*MTRR*), cytoplasmic serine hydroxymethyltransferase (*cSHMT*), dihydrofolate reductase (*DHFR*), betaine-homocysteine S-methyltransferase (*BHMT*), and thymidylate synthase (*TYMS* or TS) [6]. The TS protein catalyzes the conversion of deoxyuridine monophosphate (dUMP) to deoxythymidine monophosphate (dTMP). This process is essential for the production of thymine, a type of nucleotide required for DNA synthesis and repair [7,8]. Once synthesized, dTMP is then metabolized intracellularly to dTTP (the triphosphate form), an essential precursor for DNA biosynthesis. This reaction is also critical, as it maintains the essential metabolic requirements needed for cellular proliferation and growth. Because of its key role in DNA replication, human TS is an anticancer drug target [9,10]. In particular, this protein is targeted by the widely used anticancer agent 5-FU, which is active against solid tumors, such as breast, head and neck, and colon cancers. Notably, elevated TS levels are correlated with a poorer prognosis, and increased amounts of TS in tumors are associated with resistance to 5-FU chemotherapy [11]. Therefore, TS levels have been suggested as a prognostic factor for CRC survival [12,13] and the response of tumor cells to 5-FU therapy.

A number of studies have investigated possible roles for polymorphisms in several 1C metabolism genes in cancer, including those in TS, which may affect CRC susceptibility, disease progression, and TS-targeted chemotherapy [14,15,16,17,18]. Among these are two well-studied TS polymorphisms: (1) TS enhancer region (TSER) 2R/3R and (2) TS 1494del6 [19,20,21]. One previous study showed that CRC tumor tissue containing the triple repeat (3R) in the TS enhancer exhibits 4-fold higher TS mRNA levels than CRC tumor tissue from patients who carry the 2R variant (*p* < 0.004) [22]. This polymorphism is of clinical significance, as greater TS enzyme activity is observed in cancer cells containing the triple repeat than for those with the double repeat [19,23]. Another study reported that in 5-FU–treated patients, the presence of a homozygous 6-bp deletion (−6 bp/−6 bp) in the TS 3′-untranslated region (UTR), which decreases TS mRNA stability and is associated with decreased TS expression in vivo, is correlated with increased treatment efficacy in at least 20% of the study population [24]. In particular, despite the large amount of data showing the association between these polymorphisms and CRCs, the genetic mechanisms underpinning the role of most TS polymorphisms in CRCs are unclear. [25]. Recent studies have revealed that the 3′-UTR region serves as the binding site for micro RNAs (miRNAs), small non-coding RNAs that function to promote post-transcriptional silencing of target gene expression. Notably, there is also evidence that polymorphisms in the 3′-UTR of a number genes, which are predicted to alter binding sites for specific miRNAs, may be involved in various cancers [26,27,28,29]. However, the function of such variants in the 3′-UTR of TS remains unclear.

In the present study, we performed a database search and identified two single nucleotide polymorphisms (SNPs) in the TS 3′-UTR: TS 1100T>C (rs699517) and TS 1170A>G (rs2790). These SNPs are found at minor allele frequencies of >5% in the Asian population; however, it is not known whether they display any genetic associations with CRC or if there is variation in TS expression as a function of these 3′-UTR polymorphisms. Therefore, in this study, we investigated whether these polymorphisms in the TS 3′-UTR correlate with CRC development and TS mRNA expression levels.

## 2. Materials and Methods

### 2.1. Ethics Statement

The study protocol was approved (IRB No. 2009-08-077) by the Institutional Review Board of CHA Bundang Medical Center, Seongnam, South Korea. All study subjects provided written informed consent, and all of the methods applied in this study were carried out in accordance with the approved guidelines.

### 2.2. Subjects

We conducted a case-control study of 850 individuals, including 450 CRC patients and 400 healthy controls. A total of 450 patients diagnosed with CRC at CHA Bundang Medical Center (Seongnam, South Korea) were enrolled from June 1996 to January 2010. We included only CRC patients who had undergone surgical resection with a curative intent and who had histologically confirmed adenocarcinoma. Within the CRC cohort, 155 consecutive patients with proximal colon cancer (i.e., from the cecum to the splenic flexure), 101 consecutive patients with distal colon cancer (i.e., descending and sigmoid colon), 186 consecutive patients with rectal cancer, 4 consecutive patients with mixed colon cancer, and 4 consecutive patients with unclassified CRC underwent primary surgery. We retrospectively obtained information on the date of diagnosis, pathological stage, relapse, and death. Tumor staging of CRCs was performed according to the sixth edition of the *American Joint Committee on Cancer (AJCC) Cancer Staging Manual*.

The control group consisted of 400 individuals who were randomly selected following a health screening. We excluded patients with a history of thrombotic diseases or cancer. Patients with a high baseline blood pressure (systolic ≥ 140 mm Hg or diastolic ≥ 90 mmHg) on more than one occasion or with a history of antihypertensive medication were classified as having hypertension (HTN). Patients with high fasting plasma glucose (≥126 mg/dL), who took oral hypoglycemic agents, or with a history of insulin treatment were classified as having diabetes mellitus (DM). All study subjects were ethnic Koreans and provided written informed consent to participate.

### 2.3. Phenotype Measurements

Anthropometric measurements included the body mass index (BMI). Systolic and diastolic blood pressures were measured in the seated position after 10 min of rest. For measurements of physiological parameters, 3 mL blood samples were obtained after overnight fasting. Plasma glucose levels were measured in duplicate by the hexokinase method, adapted to an automated analyzer (TBA 200FR NEO, Toshiba Medical Systems, Tokyo, Japan). Levels of high-density lipoprotein cholesterol (HDL-C) were determined by enzymatic colorimetric methods, using commercial reagent sets (Toshiba Medical Systems). Plasma homocysteine (Hcy) concentrations were measured by fluorescent polarizing immunoassay (FPIA) with the IMx automated immunohistochemistry system (Abbott Laboratories, Chicago, IL, USA). Plasma concentrations of fatty acids (FA) were determined using a radioassay kit (ACS:180, Bayer, Tarrytown, NY, USA).

### 2.4. Genotyping

DNA was extracted from leukocytes using the G-DEX II Genomic DNA Extraction Kit (Intron Biotechnology, Seongnam, Korea), according to the manufacturer instructions. Genotyping for TS 1100T>C and TS 1170A>G was performed by polymerase chain reaction–restriction fragment length polymorphism (PCR-RFLP) assays. Primers and TaqMan probes were designed using Primer Express Software (v. 2.0) and synthesized by Applied Biosystems (Foster City, CA, USA), with the FAM and JOE reporter dyes. To detect the TS 1100T>C and 1170A>G genotypes, PCR amplification was performed with forward (5′-GGT ACA ATC CGC ATC CAA CTA TTA-3′) and reverse (5′-CTG ATA GGT CAC GGA CAG ATT T-3′) primers, producing a fragment of 170 bp. PCR products were digested with 5U *Ban*II (TS 1100T>C) or 3U *Mbo*II (TS 1170A>G) for 16 h at 37 °C.

For 1100T>C, the TT genotype was identified by restriction products of 170 bp; the TC genotype was identified by products of 170 bp, 108 bp, and 62 bp; and the CC genotype was identified by products of 108 bp and 62 bp. For 1170A>G, the AA genotype was identified by restriction products of 170 bp; the AG genotype was identified by products of 170 bp, 142 bp, and 28 bp; and the GG genotype was identified by products of 142 bp and 28 bp products. For each polymorphism, 30% of the PCR assays were randomly selected and repeated, followed by DNA sequencing with an ABI 3730xl DNA Analyzer (Applied Biosystems), to validate the experimental findings. The concordance for quality control samples was 100%.

### 2.5. Quantitative Reverse Transcription-PCR

Total RNA for quantitative reverse transcription (qRT)-PCR was extracted from 94 colorectal tissues (47 tumor and 47 tumor-adjacent tissues) of CRC patients using TRIzol reagent (Invitrogen, Thermo Fisher Scientific, Waltham, MA, USA) according to the manufacturer instructions. Synthesis of cDNA from total RNA was performed with the SuperScript III First-Strand Synthesis System (Invitrogen). TS expression levels in tissue were calculated using a comparative CT (2^−ΔΔCT^) method, with the 18s rRNA gene serving as an internal control. Primer sequences for amplification were as follows: 18s rRNA: forward 5′-AAC TTT CGA TGG TAG TCG CCG-3′ and reverse 5′-CCT TGG ATG TGG TAG CCG TTT-3′; TS: forward 5′-CAA CCC TGA CGA CAG AAG AA-3′ and reverse 5′-GCT CAC TGT TCA CCA CAT AGA-3′.

### 2.6. Statistical Analysis

To analyze baseline characteristics and compare patient and control data, chi-square tests were used for categorical data, and Student’s t-tests were used for continuous data. Associations between TS 3′-UTR polymorphisms and CRC incidence were calculated using adjusted odds ratios (AORs) and 95% confidence intervals (95% CIs) from multivariate logistic regression analysis, adjusted for age, gender, HTN, DM, BMI, and HDL-C. The variables HTN, DM, BMI, and HDL-C were selected for adjustment, since the risk factors for metabolic syndrome are closely associated with CRC [30]. Allelic gene-gene interaction analysis was performed with the open-source multidimensional reduction (MDR) software package (v. 2.0) available from www.epistasis.org. The MDR method consists of two main steps [31]. First, the best combination of multi-factors is selected, and second, genotype combinations are classified into high- and low-risk groups [31]. For a detailed discussion see Ritchie et al. [32] and Moore and William [33]. We constructed all possible allelic combinations by MDR analysis to identify combinations with strong synergy. Allelic combinations and haplotypes for multiple loci were estimated using the expectation-maximization algorithm with SNPAlyze (v. 5.1; DYNACOM Co, Ltd., Yokohama, Japan), and those having frequencies <1% were excluded from statistical analysis. Cox regression models were used to analyze the independent prognostic importance of various markers, and the results were adjusted for age, gender, tumor differentiation, tumor site, chemotherapy, and cancer stage. These calculations excluded 100 CRC patients for whom an insufficient medical history was obtained. Overall survival (OS) was defined as the time from surgery until death or the last follow-up, and relapse-free survival (RFS) was defined as the time from surgery until cancer recurrence or the last follow-up. Adjusted hazard ratios (AHRs) are presented with a 95% CI. Participants were followed for a median of 34 months (range, 4–173 months). The estimated 3-year OS and RFS rates for all patients were 82.6% and 81.7%, respectively. Analyses were performed using GraphPad Prism v. 4.0 (GraphPad Software Inc., San Diego, CA, USA) and MedCalc v. 12.7.1.0 (MedCalc Software, Mariakerke, Belgium).

## 3. Results

### 3.1. Patient Characteristics

We collected blood samples and clinical data from 450 CRC patients, including 212 men and 238 women. The mean age was 62.5 years (standard deviation (SD) = 12.29) (Table 1). A total of 260 patients (57.8%) had colon cancer, and 186 patients (41.3%) had rectal cancer. Pathological staging after curative resection was as follows: 42 (9.3%) patients with tumor node metastasis (TNM) stage I, 189 (42.0%) patients with stage II, 173 (38.4%) patients with stage III, and 46 (10.2%) patients with stage IV. CRC patients and control subjects showed no differences in age or gender (*p* = 0.162 and *p* = 0.177, respectively). However, we found that CRC patients were significantly more likely to have HTN (*p* < 0.001), DM (*p* < 0.001), low HDL-C levels (*p* < 0.001), and decreased FA levels (*p* = 0.043), relative to healthy controls.

### 3.2. Genotype Frequencies of TS 3′-UTR Variants

Table 2 shows the distributions of genotypes for TS 3′-UTR polymorphisms in CRC patients and control subjects. The genotype frequencies in both groups were consistent with expectations under Hardy–Weinberg equilibrium (HWE). In the total CRC group, TS 1170AG (AOR = 1.55, 95% CI = 1.13–2.11) and 1170GG (AOR = 3.19, 95% CI = 1.91–5.34) were found to be significantly associated with CRC susceptibility. We then divided CRC patients into two subgroups, consisting of those with colon and rectal cancer. For the colon subgroup, TS 1170AG (AOR = 1.73, 95% CI = 1.20–2.50) and 1170GG (AOR = 4.31, 95% CI = 2.42–7.66) were significantly associated with CRC susceptibility, and in the rectum subgroup, 1170GG (AOR = 2.07, 95% CI = 2.42–7.66) was significantly associated with CRC susceptibility. No statistically significant associations were found for TS1100T>C genotypes.

### 3.3. Effects of Combined Genotypes and Allelic Gene-Gene Interactions for TS 3′-UTR Variants on CRC Incident Rates

To evaluate possible combined effects between TS 3′-UTR polymorphic loci on CRC incidence, we performed logistic regression on the combined genotypes (Table 3). In the total CRC group, TS 1100TT-1170GG (AOR = 5.24; 95% CI = 2.67–10.27), 1100TC-1170AG (AOR = 3.06; 95% CI = 1.73–5.43), and 1100CC-1170AA (AOR = 3.00; 95% CI = 1.45–6.21) showed significant AOR values >3.00. In the colon and rectum subgroups, TS 1100TT-1170GG (AOR = 6.10; 95% CI = 2.81–13.23, AOR = 4.04; 95% CI = 1.69–9.69, respectively), 1100TC-1170AG (AOR = 3.09; 95% CI = 1.59–6.00, AOR = 3.43; 95% CI = 1.57–7.45, respectively), and 1100CC-1170AA (AOR = 3.12; 95% CI = 1.34–7.28, AOR = 2.98; 95% CI = 1.11–8.01, respectively) showed significant AOR values >3.00. For the TS haplotypes, in the total CRC group, TS 1100T-1170G (AOR = 2.00; 95% CI = 1.56–2.55) and 1100C-1170A (AOR = 1.51; 95% CI = 1.17–1.96) were associated with increased CRC incidence rates. In the colon and rectum subgroups, TS 1100T-1170G (AOR = 2.25; 95% CI = 1.69–3.00, AOR = 1.71; 95% CI = 1.24–2.34, respectively) and TS 1100C-1170A (AOR = 1.57; 95% CI = 1.16–2.13, AOR = 1.49; 95% CI = 1.08–2.07, respectively) were significantly associated with CRC susceptibility.

### 3.4. Stratified Effects of TS 3′-UTR Polymorphisms on CRC Incidence

To determine whether certain alleles were associated with CRC incidence in specific subsets of patients, we conducted a stratified analysis of the data according to age, gender, hypertension (HTN), diabetes mellitus (DM), tumor node metastasis (TNM) stage, body mass index (BMI), high density lipoprotein-cholesterol (HDL-C), homocysteine (Hcy), and folate (FA) (Table 4). To establish cut-off values in the ranges of Hcy and FA serum levels, we selected upper and lower 25% cut-offs for Hcy and FA, respectively. These values correspond to 11.7 μmol/L for Hcy and 4.58 ng/mL for FA. Table 4 and Table 5 contain results from our stratified analysis of constructed TS haplotypes and TS combined genotypes. For the TS 1100/1170 combined genotypes, common subset-specific associations for 1100TC-1170AG, and 1100CC-1170AA were found for the male, TNM I or II stage, HTN, Hcy < 11.7 μmol/L and FA ≥ 4.58 ng/mL subgroups; specific associations for TS 1100TT-1170GG were found for the female, TNM I or II, BMI ≥ 25 kg/m^2^, Hcy < 11.7 μmol/L, and FA ≥ 4.58 ng/mL subgroups; and specific associations for TS 1100TT-1170AG were found for the tumor size < 5 cm, TNM I or II BMI < 25 kg/m^2^, HDL-C ≥ 40 (M)/50 (F) mg/dL, Hcy < 11.7 μmol/L, and FA ≥ 4.58 ng/mL subgroups (Table 4). TS 1100C-1170A haplotypes displayed common subset-specific associations with the age ≥ 62 years, male, ≥ 5-cm tumor, TNM I or II stage, HTN, DM, BMI (<25 kg/m^2^), HDL-C ≥ 40 (male)/50 (female) mg/dL, Hcy ≥ 11.7 μmol/L and FA (≥4.58 ng/mL) subgroups (Table 5).

### 3.5. Gene-Environment Combined Effects of TS 3′-UTR Polymorphisms on CRC Incidence

Because cancer risk and prognosis are determined by a complex interplay of genetic and environmental factors, we calculated the combined gene-environment effects on CRC incidence (Appendix A). To perform this gene-environment interaction analysis for CRC incidence rates, we selected the environmental factors, such as HTN, DM, HDL-C, Hcy, and FA, which showed statistically significant differences in subjects with total CRC, colon, or rectal cancer relative to controls. Of the single genotypes, TS 1170AG+GG displayed strong gene-environment combined effects with HTN and FA < 4.58 ng/mL on CRC incidence. For the TS 1100/1170 combined genotypes, TS 1100TT-1170GG, 1100TC-1170AG, and 1100CC-1170AA showed strong gene-environment combined effects with HTN, DM, and HDL-C < 40 (male)/50 (female) mg/dL on CRC prevalence.

### 3.6. CRC Progression According to TS 3′-UTR Polymorphisms

To identify genetic associations with CRC prognosis, we conducted a survival analysis for OS and RFS according to TS 3′-UTR polymorphisms. From our total cohort, 100 CRC patients had an insufficient medical history and were therefore excluded from this analysis. Characteristics of the 350 subjects included in this survival analysis are summarized in Appendix A which show the OS and RFS rates for various TS 3′-UTR polymorphisms. From this analysis, we found a significant association between TS 1100TC+CC and poor OS (AHR = 1.82; 95% CI = 1.07–3.78) and RFS rates (AHR = 2.00; 95% CI = 1.19–3.35) (Figure 1). Appendix A presents AHR values of prognosis risk factors according to TS 3′-UTR polymorphisms. TNM stage, tumor differentiation, HDL-C, and FA significantly affected OS and RFS rate of CRC patients. The poorest AHR values for OS and RFS by HDL-C levels occurred based on the TS 1100TC-1170AG combined genotype. Of AHR values for OS and RFS by plasma FA levels, the TS 1100TC-1170AA combined genotype showed significantly higher values than others.

### 3.7. Expression of TS 3′-UTR Polymorphisms

We quantified expression of TS in tissue samples and determined whether we could detect differences in expression based on the TS 3′-UTR polymorphisms and tested haplotypes (Figure 2, Appendix A). Compared with the wild-type (WT) 1100TT genotype, expression of 1100TC and 1100CC was found to be significantly increased in tumor tissue. In the case of TS 1170A>G, we detected increased expression of 1170AG and 1170GG relative to 1170AA (WT) in tumors, similar to what we observed for the TS 1100T>C genotype. In addition, we found that the TS 1100T-1170G and 1100C-1170A haplotypes were associated with increased TS expression. Thus, our data indicate that expression of the mutant type was significantly increased in CRC for both TS 1100 and TS 1170.

### 3.8. Expression of Target miRNAs

We next quantified expression of *miR-203* and *miR-124-1* in tissue samples and determined whether we could detect differences in expression, based on the TS 3′-UTR polymorphisms and haplotypes (Appendix A). Compared with the wild-type (WT) 1100TT genotype, expression of *miR-203* in 1100TC and 1100CC was found to be decreased in tumor tissue. In the case of TS 1170A>G, we detected decreased expression of 1170AG and 1170GG relative to 1170AA (WT) in tumors, similar to what we observed for the TS 1100T>C genotype. However, no statistically significant inverse correlation has been identified in the expression between TS and miRNA.

## 4. Discussion

In the present study, we investigated the association between two new 3′-UTR polymorphisms in the TS gene, 1100T>C (rs699517) and 1170A>G (rs2790), and the occurrence and prognosis of CRC. We found that the combined genotype and haplotype of TS 1100T>C and 1170A>G were strongly associated with CRC susceptibility. For associations with prognosis factors (3-year OS and RFS rates), TS 1100T>C was an effective predictor for poor prognosis. We also found that TS 1170A>G genotypes are strongly associated with CRC susceptibility, with an increased risk of CRC observed in those with the mutant 1170G allele compared to the WT 1170A allele. Our data further indicate that the TS 1100TC, TS 1100CC, and TS 1170AG genotypes, and the TS 1100T-1170G and 1100C-1170A haplotypes are associated with elevated TS expression in CRC.

Previous studies have focused on known *TYMS* polymorphisms in the promoter region, including the TSER 2R/3R repeat and 5′-UTR (rs45445694), as well as a 6-bp in/del of 3′-UTR (TS 1494del6b, (rs16430)) [22]. In particular, TS 3R alleles, which promote elevated TS expression, have been shown to be potent genetic risk factors for increased CRC susceptibility in meta-analyses [34,35,36]. Critically, an understanding of how elevated levels of TS promote carcinogenic events could also help to explain why TS 3′-UTR polymorphisms may affect CRC occurrence and prognosis. According to a recent report, compared with the rs2790 AA genotype at the 3′-UTR of TS, the rs2790 GG genotype was associated with a significantly higher risk of acute lymphoblastic leukaemia (ALL) [37]. The present study confirmed that the presence of the TS 3′UTR polymorphism (rs151264360) increased the risk of persistence of cervical cancer [38]. Another report shows that the rs2790 AA genotype has a higher risk of death than GA or GG genotypes in Chinese patients with non-small cell lung cancer (NSCLC) [39].

Plasma FA concentrations inversely correlate with Hcy levels in vivo [40], and it has previously been shown that elevated TS expression can promote increased Hcy and decreased FA levels, leading to induction of CRC development [41,42]. In particular, depletion of FA is thought to promote colorectal carcinogenesis via the induction of DNA breaks in the chromosome [43]. There are plausible mechanisms by which FA deficiency may create such breaks, including uracil misincorporation and impaired DNA repair [44,45]. FA deficiency reduces deoxythymidylate synthesis from deoxyuridylate, and the resulting nucleotide imbalance accelerates the incorporation of uracil bases into DNA. Uracil in DNA is excised by a repair glycosylase, and in the process, a transient single-strand break is generated in the chromosome [46]. FA supplementation has been proposed as a possible solution, however, animal studies have shown that this may increase CRC risk and accelerate CRC progression if too much is given, or if it is provided after neoplastic foci are established in the colorectum [47,48]. CRC incidence has been influenced by a variety of risk factors. Therefore, stratified analyses were useful in elucidating CRC epidemiology resulting from a diversity of confounding variables. Our data show that common subset-specific associations for 1100TC-1170AG, 1100CC-1170AA and TS 1100TT-1170GG were found for the gender, TNM I or II stage, HTN, Hcy and FA subgroups; TS 1100C-1170A haplotypes displayed common subset-specific associations with the age, gender (male), tumor size, TNM I or II stage, HTN, DM, BMI (<25 kg/m^2^), HDL-C ≥ 40 (male)/50 (female) mg/dL, Hcy (≥11.7 μmol/L) and FA (≥4.58 ng/mL) subgroups. Because cancer risk is determined by the complex interplay of genetic and environmental factors, we calculated combined gene-environment effects on CRC susceptibility. Notably, our data show that when compared with TS 1170AA WT, the TS 1170AG+GG genotypes display strong gene-environment combined effects with HTN and FA < 4.58 ng/mL on CRC incidence. We did not observe any significant gene-environment combined effects with Hcy concentrations, consistent with previous studies that did not detect a significantly elevated CRC risk with increasing plasma Hcy levels [49,50]. Previous studies on the association between plasma Hcy levels and risk of colorectal neoplasia have further yielded complicated results.

Here, we detected increased TS expression in subjects with the TS 1100TC and CC and the TS 1170AG and GG genotypes, relative to those with the respective WT genotypes. Additionally, we find that the risk of CRC is increased in individuals with the mutant (1100CC, 1170GG) or heterozygous genotypes (1100TC, 1170AG) compared to the WT 1100TT and 1170AA genotype. Notably, elevated TS levels are correlated with a poorer prognosis, and increased amounts of TS in tumors are associated with resistance to 5-FU chemotherapy. Our data indicate that expression of the mutant type was significantly increased in CRC for both TS 1100, TS 1170 and haplotypes. 5-Fluorouracil (5-FU) is a widely used chemical agent for treating colorectal cancer [51]; however, resistance to 5-FU can decrease its therapeutic efficacy. Previous reports have suggested that miRNAs might represent an effective strategy for preventing resistance and improving treatment outcomes. For example, it was shown that *miR-203* inhibition enhances cellular resistance to 5-FU, whereas *miR-203* overexpression increases 5-FU sensitivity [52]. Notably, functional experiments further showed that the inhibitory effect of *miR-203* on tumor growth under 5-FU exposure is mediated by downregulation of TS, which increases cellular sensitivity to 5-FU. Another report provided direct evidence that *miR-192* and *miR-215* also influence sensitivity to 5-FU treatment [53]. These data therefore suggest that miRNAs may be useful biomarkers for predicting 5-FU treatment outcomes. Our data show that *miR-203* concentrations inversely correlate with TS 1100T>C and TS 1170A>G levels in tissues. Genetic variation in the 3′-UTR region could affect mRNA stability and translation through an altered miRNA binding affinity. Currently, however, there are no data to directly show that miRNA binding activity is affected by TS 1100 or TS 1170 3′-UTR polymorphisms.

The reason for the observed decrease in expression of miRNAs involved in the regulation of a drug-target gene in tumor tissue remains unclear. However, a possible explanation can be found in a previous report demonstrating that the loss of the tumor suppressor PKCz in CRC cells is associated with lower ADAR2 activity. This promotes the loading of *miR-200s* into extracellular vesicles (EVs), thereby decreasing their intracellular steady-state levels. Loss of this axis induces the epithelial-to-mesenchymal transition (EMT) and increased liver metastases, which can be inhibited in vivo by blocking *miR-200* release. Therefore, the PKCz/ADAR2 axis is a critical regulator of CRC metastases through modulation of *miR-200* levels [54]. Further studies, however, are needed to determine whether a similar mechanism regulates the steady-state levels of *microRNAs*. Specifically, it will be necessary to identify the factors that regulate expression of this subset of miRNAs. In addition, it is important to determine how regulation of the 3′-UTR of TS by *microRNAs* might influence 5-FU treatment, and whether this also affects cell proliferation and cancer progression. Importantly, these studies also have the potential to clinically impact many diseases related to 1C metabolism.

We could not find differences in miRNA binding activity or pre-miRNA expression of target miRNAs as a function of genotype at these loci. Despite a lack of data, these miRNAs may be important genetic factors in the prevalence and progression of CRC because their expression is altered in some of the genotypes and haplotypes tested here. Further studies are needed to directly test for miRNA binding activity to TS 3′-UTR polymorphisms, to determine the mechanism by which these polymorphisms may influence cellular proliferation and cancer progression.

This study has several limitations. First, the manner in which 3′-UTR polymorphisms in the TS genes affect development of CRC is still unclear. Second, the lack of information regarding additional environmental risk factors in CRC patients remains to be investigated. Lastly, the population of this study was restricted to patients of Korean ethnicity. Although results from our study provide the first evidence for 3′-UTR variants in TS genes as potential biomarkers of CRC prevention and prognosis, a prospective study on a larger cohort of patients is warranted to validate these findings.

In conclusion, this study is the first to identify an association between TS 3′-UTR polymorphisms (rs699517, rs2790) and an increased risk of CRC. We found that TS 1170A>G genotypes, as well as the TS 1100T-1170G and 1100C-1170A haplotypes, are strongly associated with CRC. The TS 1100TC+CC type was associated with a poor survival (OS and RFS) rate. Stratified analyses indicated that subset-specific associations for TS 1100/1170 combined genotypes were found for the age, TNM I or II stage, HTN, and FA subgroups. TS 1100C-1170A haplotypes displayed common subset-specific associations with the age, gender (male), tumor size, TNM I or II stage, HTN, DM, HDL-C and Hcy subgroups.

## Figures and Tables

**Figure 1 jpm-11-00537-f001:**
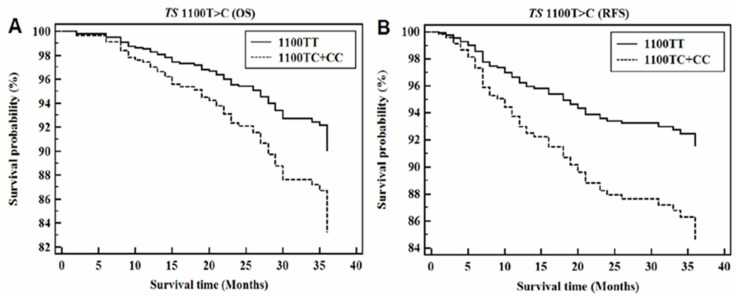
Cox proportional-hazard regression analysis for TS 1100T>C. (**A**) Overall survival (OS) curve for TS 1100T>C (Dominant model; AHR = 1.82; 95% CI = 1.07–3.08; *p* = 0.027). (**B**) Relapse-free survival (RFS) curve for TS 1100T>C (Dominant model; AHR = 2.00; 95% CI = 1.19–3.35; *p* = 0.009).

**Figure 2 jpm-11-00537-f002:**
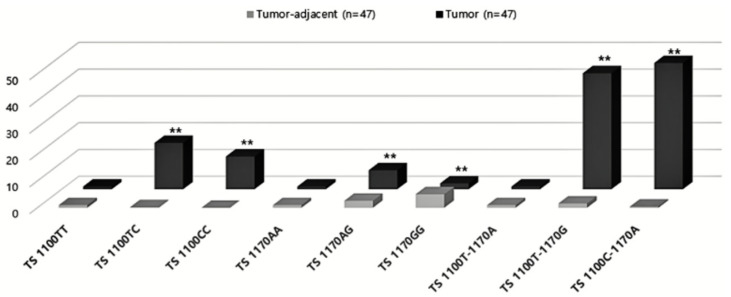
Expression of thymidylate synthase (TS) 1100T>C (rs699517) and 1170A>G(rs2790). Single nucleotide polymorphisms (rs699517 and rs2790) in the 3′-untranslated region (UTR) of TS. Quantitative reverse transcription PCR (qRT-PCR) to measure levels of TS 1100T and 1100C, TS 1170A and TS 1170G, TS 1100T-1170G and TS 1100C-1170A expression in CRC tissue. Means and individual values from three biological replicates are shown. Source data are provided in the source Appendix A. The housekeeping gene GADPH served as the loading control. ** indicates *p* < 0.05.

**Table 1 jpm-11-00537-t001:** Baseline characteristics of colorectal cancer patients and control subjects.

Characteristics	Control	CRC	*p*	Colon	*p*	Rectum	*p*
*N*	400	450		260		186	
Age: years (mean ± SD)	60.89 ± 11.72	62.05 ± 12.29	0.162	61.67 ± 12.86	0.426	62.33 ± 11.46	0.165
Gender male: *n* (%)	170 (42.5)	212 (47.1)	0.177	122 (46.9)	0.299	88 (47.3)	0.316
Hypertension: *n* (%)	157 (39.3)	279 (62.0)	<0.001	155 (59.6)	<0.001	120 (64.5)	<0.001
Diabetes mellitus: *n* (%)	166 (41.5)	253 (56.2)	<0.001	147 (56.5)	<0.001	104 (55.9)	0.002
BMI ≥ 25 kg/m^2^: *n* (%)	93 (23.3)	116 (25.8)	0.393	63 (24.2)	0.845	51 (27.4)	0.323
HDL-C < 40 (male) or 50 (female) mg/dL: *n* (%)	78 (19.5)	197 (43.8)	<0.001	111 (42.7)	<0.001	84 (45.2)	<0.001
Homocysteine: μmol/L (*n*)	9.80 ± 4.17 (395)	10.51 ± 7.76 (383)	0.115	10.26 ± 8.16 (220)	0.362	10.77 ± 7.24 (159)	0.050
Folate: ng/mL (*n*)	8.85 ± 8.06 (392)	7.77 ± 6.65 (381)	0.043	8.06 ± 7.19 (220)	0.226	7.35 ± 5.85 (157)	0.034
Tumor size: *n* (%)							
<5 cm	-	181 (40.2)	NA	92 (35.4)	NA	87 (46.8)	NA
≥5 cm	-	269 (59.8)	NA	168 (64.6)	NA	99 (53.2)	NA
TNM stage: *n* (%)							
I	-	42 (9.3)	NA	22 (8.5)	NA	20 (10.8)	NA
II	-	189 (42.0)	NA	115 (44.2)	NA	71 (38.2)	NA
III	-	173 (38.4)	NA	94 (36.2)	NA	79 (42.5)	NA
IV	-	46 (10.2)	NA	29 (11.2)	NA	16 (8.6)	NA

Abbreviations: CRC, colorectal cancer; SD, standard deviation; BMI, body mass index; HDL-C, high density lipoprotein-cholesterol; *MTHFR*, methylenetetrahydrofolate reductase; TNM, tumor node metastasis. *p*-values were calculated by chi-square test for categorical data and two-sided t-test for continuous data. NA: Statistics cannot be processed because there is no comparison group.

**Table 2 jpm-11-00537-t002:** Genotype frequencies of TS 3′-UTR polymorphisms between CRC patients and control subjects.

Genotypes	Control (*n* = 400)	CRC (*n* = 450)	AOR (95% CI)	*p*	Colon (*n* = 260)	AOR (95% CI)	*p*	Rectum (*n* = 186)	AOR (95% CI)	*p*
TS 1100T>C										
TT	195 (48.8)	223 (49.6)	1.00 (ref)	NA	134 (51.5)	1.00 (ref)	NA	86 (46.2)	1.00 (ref)	NA
TC	177 (44.3)	189 (42.0)	0.99 (0.73–1.33)	0.922	105 (40.4)	0.88 (0.62–1.25)	0.482	84 (45.2)	1.21 (0.82–1.78)	0.345
CC	28 (7.0)	38 (8.4)	1.27 (0.73–2.22)	0.401	21 (8.1)	1.25 (0.66–2.37)	0.489	16 (8.6)	1.28 (0.62–2.64)	0.511
TS 1170A>G										
AA	195 (48.8)	155 (34.4)	1.00 (ref)	NA	82 (31.5)	1.00 (ref)	NA	71 (38.2)	1.00 (ref)	NA
AG	176 (44.0)	224 (49.8)	1.55 (1.13–2.11)	0.006	132 (50.8)	1.73 (1.20–2.50)	0.003	90 (48.4)	1.42 (0.95–2.11)	0.085
GG	29 (7.3)	71 (15.8)	3.19 (1.91–5.34)	<0.001	46 (17.7)	4.31 (2.42–7.66)	<0.001	25 (13.4)	2.07 (1.08–3.95)	0.028

Abbreviations: MTHFR, methylenetetrahydrofolate reductase; TS, thymidylate synthase; CRC, colorectal cancer; AOR, adjusted odds ratio (adjusted by age, gender, hypertension, diabetes mellitus, body mass index, and high density lipoprotein-cholesterol); CI, confidence interval. NA: Statistical processing was not possible because the corresponding value was a reference.

**Table 3 jpm-11-00537-t003:** Combined genotype and haplotype frequencies of TS 3′-UTR polymorphisms between CRC patients and control subjects.

Genotypes	Control (*n* = 400)	CRC (*n* = 450)	AOR (95% CI)	*p*	Colon (*n* = 260)	AOR (95% CI)	*p*	Rectum (*n* = 186)	AOR (95% CI)	*p*
TS Genotype										
1100TT-1170AA	63 (15.8)	39 (8.7)	1.00 (ref)	NA	23 (8.8)	1.00 (ref)	NA	15 (8.1)	1.00 (ref)	NA
1100TT-1170AG	103 (25.8)	113 (25.1)	1.81 (1.07–3.06)	0.027	65 (25.0)	1.75 (0.95–3.25)	0.074	46 (24.7)	1.94 (0.96–3.93)	0.067
1100TT-1170GG	29 (7.3)	71 (15.8)	5.24 (2.67–10.27)	<0.001	46 (17.7)	6.10 (2.81–13.23)	<0.001	25 (13.4)	4.04 (1.69–9.69)	0.002
1100TC-1170AA	104 (26.0)	78 (17.3)	1.40 (0.82–2.40)	0.220	38 (14.6)	1.10 (0.57–2.12)	0.785	40 (21.5)	1.85 (0.90–3.79)	0.095
1100TC-1170AG	73 (18.3)	111 (24.7)	3.06 (1.73–5.43)	<0.001	67 (25.8)	3.09 (1.59–6.00)	<0.001	44 (23.7)	3.43 (1.57–7.45)	0.002
1100CC-1170AA	28 (7.0)	38 (8.4)	3.00 (1.45–6.21)	0.003	21 (8.1)	3.12 (1.34–7.28)	0.009	16 (8.6)	2.98 (1.11–8.01)	0.030
TS haplotype										
1100T-1170A	333 (41.6)	269 (29.9)	1.00 (ref)	NA	149 (28.7)	1.00 (ref)	NA	116 (31.2)	1.00 (ref)	NA
1100T-1170G	234 (29.3)	366 (40.7)	2.00 (1.56–2.55)	<0.001	224 (43.1)	2.25 (1.69–3.00)	<0.001	140 (37.6)	1.71 (1.24–2.34)	<0.001
1100C-1170A	233 (29.1)	265 (29.4)	1.51 (1.17–1.96)	0.002	147 (28.3)	1.57 (1.16–2.13)	0.004	116 (31.2)	1.49 (1.08–2.07)	0.017

Adjusted by age, gender, hypertension, diabetes mellitus, body mass index, and high density lipoprotein-cholesterol. NA: Statistical processing was not possible because the corresponding value was a reference.

**Table 4 jpm-11-00537-t004:** Analysis of CRC incidence and association in specific patient subsets.

	TS 1100TT-1170AG		TS 1100TT-1170GG		TS 1100TC-1170AA		TS 1100TC-1170AG		TS 1100CC-1170AA	
Subsets	AOR (95% CI)	*p*	AOR (95% CI)	*p*	AOR (95% CI)	*p*	AOR (95% CI)	*p*	AOR (95% CI)	*p*
Age < 62 years	2.08 (0.98–4.43)	0.058	3.38 (1.34–8.52)	0.010	1.07 (0.50–2.32)	0.856	2.78 (1.24–6.24)	0.013	1.48 (0.53–4.14)	0.454
≥62 years	1.79 (0.85–3.77)	0.126	7.62 (2.68–21.68)	<0.001	1.87 (0.85–4.12)	0.122	3.08 (1.35–7.04)	0.008	6.63 (2.10–20.97)	0.001
Gender Male	2.12 (0.90–5.01)	0.086	4.61 (1.80–11.78)	0.001	1.94 (0.80–4.67)	0.142	3.57 (1.51–8.43)	0.004	3.45 (1.16–10.29)	0.026
Female	1.62 (0.82–3.18)	0.162	6.23 (2.24–17.34)	<0.001	1.02 (0.50–2.07)	0.953	2.70 (1.21–6.04)	0.015	2.91 (1.06–8.01)	0.039
TS < 5 cm	2.13 (1.06–4.27)	0.034	5.83 (2.37–14.32)	<0.001	1.71 (0.82–3.57)	0.154	3.88 (1.74–8.65)	<0.001	2.41 (0.83–7.00)	0.106
≥5 cm	1.65 (0.88–3.09)	0.120	5.03 (2.35–10.80)	<0.001	1.24 (0.65–2.37)	0.506	2.97 (1.53–5.77)	0.001	3.71 (1.61–8.54)	0.002
TNM I + II	2.20 (1.13–4.31)	0.021	6.24 (2.74–14.18)	<0.001	1.63 (0.81–3.25)	0.169	3.87 (1.89–7.90)	<0.001	3.35 (1.33–8.43)	0.010
III + IV	1.53 (0.81–2.89)	0.194	4.56 (2.01–10.36)	<0.001	1.22 (0.62–2.37)	0.564	2.62 (1.28–5.34)	0.008	3.00 (1.23–7.29)	0.015
HTN No	1.40 (0.66–3.00)	0.379	4.88 (1.87–12.78)	0.001	1.13 (0.53–2.44)	0.752	2.60 (1.17–5.79)	0.019	2.07 (0.73–5.85)	0.172
Yes	2.43 (1.16–5.11)	0.019	5.61 (2.10–14.96)	<0.001	2.00 (0.91–4.36)	0.083	3.86 (1.64–9.07)	0.002	4.59 (1.58–13.31)	0.005
DM No	3.29 (1.38–7.84)	0.007	5.93 (2.27–15.50)	<0.001	2.44 (0.96–6.20)	0.061	3.56 (1.43–8.86)	0.006	2.09 (0.74–5.95)	0.167
Yes	1.11 (0.55–2.23)	0.773	4.60 (1.70–12.43)	0.003	0.93 (0.45–1.92)	0.840	2.62 (1.22–5.65)	0.014	3.85 (1.28–11.54)	0.016
BMI < 25 kg/m^2^	2.13 (1.13–3.99)	0.019	4.96 (2.27–10.84)	<0.001	1.27 (0.67–2.41)	0.470	2.67 (1.39–5.12)	0.003	3.96 (1.67–9.39)	0.002
≥25 kg/m^2^	1.26 (0.48–3.33)	0.643	6.41 (1.41–29.18)	0.016	1.85 (0.66–5.17)	0.240	5.07 (1.41–18.22)	0.013	1.57 (0.35–7.08)	0.558
HDL-C < 40 (M)/50 (F)	1.37 (0.57–3.26)	0.479	2.82 (0.90–8.89)	0.077	0.96 (0.38–2.45)	0.934	3.71 (1.33–10.39)	0.013	1.44 (0.35–5.98)	0.618
≥40 (M)/50 (F)	2.12 (1.07–4.20)	0.031	7.69 (3.21–18.44)	<0.001	1.64 (0.82–3.29)	0.163	3.02 (1.48–6.16)	0.002	3.60 (1.50–8.66)	0.004
Hcy < 11.7 μmol/L	2.11 (1.10–4.07)	0.025	5.45 (2.40–12.35)	<0.001	1.41 (0.73–2.72)	0.308	2.88 (1.40–5.96)	0.004	3.14 (1.25–7.89)	0.015
≥11.7 μmol/L	0.97 (0.29–3.20)	0.957	4.14 (0.96–17.85)	0.057	1.87 (0.53–6.55)	0.327	2.88 (0.82–10.14)	0.100	4.12 (0.73–23.18)	0.108
FA < 4.58 ng/mL	1.52 (0.47–4.86)	0.485	3.31 (0.59–18.52)	0.173	0.92 (0.27–3.08)	0.888	2.44 (0.71–8.31)	0.156	3.47 (0.55–21.80)	0.184
≥4.58 ng/mL	2.00 (1.02–3.92)	0.043	8.36 (3.48–20.08)	<0.001	1.53 (0.77–3.04)	0.228	3.43 (1.62–7.27)	0.001	3.84 (1.43–10.35)	0.008

TS, Tumor size; TNM, Tumor Node Metastasis stage; HTN, hypertension; DM, diabetes mellitus; BMI, body mass index; HDL-C, high density lipoprotein-cholesterol; M, male; F, female; Hcy, homocysteine; FA, folate.11.7 was upper 25% cut-off value of Hcy in total participants. 4.58 was lower 25% cut-off value of FA in total participants. Adjusted by age, gender, HTN, DM, BMI and HDL-C.

**Table 5 jpm-11-00537-t005:** Analysis of CRC incidence and association in haplotypes of TS 1100/1170.

	1100T-1170G		1100C-1170A	
Subsets	AOR (95% CI)	*p*	AOR (95% CI)	*p*
Age < 62 years	1.91 (1.35–2.72)	<0.001	1.22 (0.85–1.76)	0.274
≥62 years	1.95 (1.37–2.79)	<0.001	1.75 (1.20–2.53)	0.003
Gender Male	2.08 (1.43–3.02)	<0.001	1.72 (1.16–2.56)	0.007
Female	1.91 (1.37–2.68)	<0.001	1.37 (0.97–1.93)	0.074
Tumor size < 5 cm	1.86 (1.36–2.53)	<0.001	1.32 (0.95–1.84)	0.103
≥5 cm	2.10 (1.57–2.81)	<0.001	1.70 (1.26–2.30)	<0.001
TNM stage I + II	2.15 (1.60–2.90)	<0.001	1.57 (1.15–2.15)	0.005
III + IV	1.82 (1.35–2.46)	<0.001	1.47 (1.07–2.02)	0.017
HTN No	1.99 (1.40–2.83)	<0.001	1.43 (0.99–2.07)	0.059
Yes	1.97 (1.39–2.80)	<0.001	1.64 (1.14–2.35)	0.008
DM No	2.04 (1.44–2.90)	<0.001	1.27 (0.87–1.84)	0.214
Yes	1.90 (1.34–2.70)	<0.001	1.77 (1.23–2.54)	0.002
BMI < 25 kg/m^2^	2.00 (1.51–2.67)	<0.001	1.48 (1.10–1.99)	0.009
≥25 kg/m^2^	1.97 (1.20–3.23)	0.008	1.66 (0.98–2.82)	0.061
HDL-C < 40 (M)/50 (F) mg/dL	1.84 (1.17–2.89)	0.008	1.35 (0.83–2.19)	0.221
≥40 (M)/50 (F) mg/dL	2.09 (1.55–2.81)	<0.001	1.56 (1.15–2.12)	0.005
Hcy < 11.7 μmol/L	1.97 (1.46–2.66)	<0.001	1.35 (0.99–1.83)	0.060
≥11.7 μmol/L	1.80 (1.05–3.08)	0.031	1.92 (1.08–3.43)	0.027
FA < 4.58 ng/mL	1.75 (1.00–3.07)	0.049	1.52 (0.87–2.65)	0.141
≥4.58 ng/mL	2.12 (1.57–2.86)	<0.001	1.50 (1.09–2.06)	0.013

HTN, hypertension; DM, diabetes mellitus; BMI, body mass index; HDL-C, high density lipoprotein-cholesterol; M, male; F, female; Hcy, homocysteine; FA, folate. The upper 25% cut-off value of Hcy in total participants was 11.7. The lower 25% cut-off value of FA in total participants was 4.58.

## Data Availability

The data presented in this study are available on request from the corresponding author.

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
