# Peer review of "3′-UTR Polymorphisms in Thymidylate Synthase with Colorectal Cancer Prevalence and Prognosis"

_jpm, 2021, doi:10.3390/jpm11060537_

Round 1

Reviewer 1 Report

This study provides evidence for the correlation between the 3'-UTR polymorphism of thymine synthase and the prognosis in colorectal cancer. TS 1170A>G genotypes, TS 1100T-1170G, and TS 1100C-1170A haplotypes, were strongly associated with elevated TS expression in CRC. TS 1100TC+CC was associated with poor survival rate. These data indicate that the 3'-UTR polymorphism in thymidylate synthase may be a prognostic factor for CRC survival.

Suggestion:

  1. What does 2.60 on line 72 mean?
  2. In lane 198, there were 450 cases of CRC, including 260 cases of colon cancer and 186 cases of rectal cancer. What is the diagnosis of the other 4 patients?
  3. In recent years, some studies have investigated the relationship between the 3'-UTR polymorphism of thymine synthase and cancer. It is recommended to compare and discuss recent relevant results.
  4. The display form of the table can be modified and expressed more clearly, because the original format is not convenient for readers to obtain information clearly.

Reviewer 2 Report

3’UTR Polymorphism in Thymidylate Synthase with Colorectal Cancer Prevalence and Prognosis-review

This current article investigated the association between two new 3’UTR polymorphisms in the TS (thymidylate Synthase)gene, 1100T>C and 1170A>G and the occurrence and prognosis of CRC. As we all know Colorectal cancer (CRC) is the third most common cancer worldwide with a variable incidence and several intrinsic factors, such as age, sex, diabetes mellitus, obesity, and inflammatory bowel disease, as well as extrinsic variables including cigarette smoking, inadequate fiber intake, high alcohol consumption and red meat consumption. It is important to have a lot of informations about tumorigenesis and about risk factors among this type of patients. One-carbon (1C) metabolism has received a crucial attention for its role in cancer malignancies and consequently therapeutic agents targeting this pathway are increasing being developed as anticancer agents. The TS protein catalyzes the conversion of deoxyuridinemonophosphate to deoxythymidine monophostphate; process that is indispensable for the production of thymine, a nucleotide neede for DNA synthesis and repair. Increased TS expression in subjects with the TS 1100TC and CC and the TS 1170AG and GG genotypes, relative to those with respective WT (wild type) genotype is an important statement in treatment of CRC. Combined genotype and haplotype of TS 1100T>C and 1170A>G were strongly associated with CRC susceptibility and for associations with prognosis factor (3-year OS and RFS rates), TS 1100T>C was an effective predictor for poor prognosis and increased amounts of TS in tumors are associated with resistance to 5-FU chemotherapy, a widely chemical agent used for treating colorectal cancer. Some reports suggested that miRNA might represent an effective strategy for preventing resistance and improving treatments outcomes. MiRNA may be an important genetic factor to the prevalence and progression of CRC because their expression is altered in some of the genotypes and haplotypes as you demonstrated in your study, and it is important todirectly test for miRNA binding activity to TS 3’-UTR polymorphisms to determine the mechanism by which these polymorphisms may influence cellular proliferation and cancer progression.

This type of study is one with huge perspectives among the management of CRC and we need to have a particular therapy based on genotypes modifications.

This present article is written in a clear and concise manner, with precision and clarity, proving to be valuable asset in it’sfield.

Round 2

Reviewer 1 Report

Dear Author

This study provides important evidence for the correlation between 3'-UTR polymorphism of thymine synthase and the prognosis in colorectal cancer. But the table display format of this article has not been modified. It is not a modification of the table title, but the presentation of the content data. For example, the same parameter must have a consistent representation format, one p value will not be displayed in two rows, and the 95% CI should be listed in the same row, etc. So that readers can clearly obtain the result information.

In your reply to point 3, the newly added references in the discussion are 38, 39, 40, but not 53, 54, 55. And there are repeated sentences in this newly added discussion paragraph.

Author Response

May, 23, 2021

Title: Association between 3'-UTR Polymorphisms in Thymidylate Synthase and Colorectal Cancer Prevalence and Prognosis

Dear Editor-in-Chief:

We would like to thank you for the 2nd revision letter and the opportunity to resubmit a revised copy of this manuscript (jpm-1218599)

We have made the necessary corrections in accordance with the suggestions of the reviewer. We hope you will find our revised manuscript acceptable for publication in the Journal of Personalized Medicine

Sincerely,

Nam Keun Kim, Ph.D.

Institute for Clinical Research, Bundang CHA General Hospital, School of Medicine, CHA University, 351 Yatap-dong, Bundang-gu, Seongnam 463-712, South Korea

Tel: 031-780-5762, Fax: 031-780-5766, E-mail: nkkim@cha.ac.kr
